# The optimization of electrochemical hydride generation technology for treating antimony-containing wastewater

Jingjing Chen[1], Guoping Zhang[2], Yi Bai[1], Yan Xiong[3], Xing Liu[1]*

1 School of Geography and Resources, Guizhou Education University, Guiyang, China, 2 State Key Laboratory of Environmental Geochemistry, Institute of Geochemistry, Chinese Academy of Sciences, Guiyang, China, 3 College of Chemistry and Materials Engineering, Guiyang University, Guiyang, China

* liux@gznc.edu.cn

## Abstract

Antimony (Sb) is extensively utilized in industrial activities, but most of its compounds exhibit human toxicity and are classified as priority-controlled pollutants. Unlike traditional electrochemical methods that remove metallic pollutants via coagulation or precipitation, electrochemical hydride generation technology converts antimony (Sb) in wastewater into stibine gas ($SbH_3$) for efficient removal. Furthermore, the generated $SbH_3$ can be decomposed thermally to partially recover metallic antimony. In synthetic wastewater treatment (Sb = 5 mg/L), the proton exchange membrane (Nafion117) electrolysis device achieved an antimony removal efficiency of 72.8 ± 2.2%, outperforming traditional cation-exchange membranes. This enhancement is attributed to the membrane's proton-selective transport and high H conductivity. Increasing the temperature enhanced the generation and release of $SbH_3$, with the higher removal efficiency of 87.3 ± 2.6% achieved at approximately 30 °C. However, temperatures exceeding 30 °C could lead to the partial decomposition of $SbH_3$ back into the solution, thereby affecting removal efficiency. Ultrasonic stirring in the cathode chamber significantly enhanced Sb removal from high-concentration solutions (5 mg/L), while magnetic stirring was more suitable for lower-concentration solutions. Orthogonal experiments revealed that due to the competitive relationship between hydrogen generation and $SbH_3$ generation, as well as the gas-blocking effect, current intensity and electrode area both had a significant impact on Sb removal. Under appropriate current intensity and electrode area conditions (0.5 A, 20 cm²), a high removal rate of 78.5 ± 4.6% can be achieved. Consequently, employing a Nafion membrane coupled with ultrasonic agitation under optimized conditions (30°C, 25 mA/cm²) effectively accelerates antimony removal kinetics and enhances elimination efficiency. However, the substantial reduction in current efficiency and elevated energy consumption induced by competitive hydrogen evolution represent critical challenges requiring urgent resolution. This treatment approach provides a technical

**Data availability statement:** All relevant data are within the manuscript and its Supporting Information files.

**Funding:** Guizhou Provincial Science and Technology Projects (No. ZK [2022] 328), Natural Science Foundation of Education Department of Guizhou Province (No. KY [2022] 300), Guizhou Science Association Pioneering Project (No. 2025XZQYXM-01-06), Natural Science Foundation of Guizhou Education University (No. 2021BS017). The funders had no role in study design, data collection and analysis, decision to publish, or preparation of the manuscript.

**Competing interests:** The authors have declared that no competing interests exist.

reference for shifting from mere contaminant removal to resource recovery. The integration of removal and recovery processes holds substantial potential for implementing circular economy models in mining and metallurgical industries.

## 1. Introduction

Antimony (Sb) is an important non-ferrous metal widely used in industry such as flame retardants, batteries, plastics, paints, glass, and alloys [1,2]. The global Sb consumption exceeding 20,000 tons annually (USGS Mineral Commodity Summaries 2025), has led to widespread environmental contamination, particularly through mining effluents and electronic waste leaching [3–5]. Studies have shown that the average Sb content in water bodies around the Beaver Brook antimony mine tailings area in Newfoundland, Canada is 9400 µg/L, with a maximum of 26400 µg/L [6]. Another study investigated Sb contamination in the Lianxi River, located near China's largest antimony mine in Xikuangshan, Hunan. The results revealed that the Sb content ranged from 942 to 2134 µg/L, with an average of 1373 µg/L. Severe pollution indirectly led to the Sb content in nearby drinking water sources reaching up to 415 µg/L [7]. In the Banpo Sb mine in Dushan County, Guizhou Province, China, the Sb concentrations in the pore water of mill tailing slurry and adit water were as high as 7.9 and 1.3 mg/L, respectively [8]. In water bodies, Sb mainly exists in two oxidation states, Sb(III) and Sb(V), which are influenced by the redox conditions of the water environment. Generally, Sb(V) mainly exists as $Sb(OH)_6^-$, while Sb(III) exists as $Sb(OH)^{2+}$, $SbO^+$, $SbO_2^-$, etc. [9]. In terms of toxicity, Sb(III) is approximately ten times more toxic than Sb(V) [10]. Most Sb compounds exhibit toxicity to humans by damaging nervous/ enzymatic systems, with Sb concentrations exceeding 5 µg/L (EU limit) [11,12]. As a priority-controlled pollutant designated by both the US Environmental Protection Agency (EPA) and European Union (EU) [1,13], Sb requires urgent remediation efforts to meet the <Water Framework Directive> and <Clean Water Act> standards.

Currently, methods for treating antimony pollution include adsorption, membrane separation, ion exchange, etc. [14,15]. The adsorption process is simple to operate and offers cost-effectiveness advantages [16]. Previous studies have demonstrated that biochar composites, loaded with nano-zero-valent bimetal (iron/copper) and prepared using the $NaBH_4$ liquid-phase reduction method, exhibit strong adsorption capacities for Sb(III), with a maximum adsorption capacity of up to 50.96 mg/g [17]. Membrane separation methods, characterized by a small footprint, low energy consumption, and high efficiency, can achieve Sb removal rates exceeding 95% through reverse osmosis by adjusting the ion concentration gradient in the solution [18]. The ion exchange method is noted for its high selectivity and removal efficiency. Studies have applied this method to treat mixed wastewater from antimony mining and smelting operations, reducing arsenic concentrations to below industrial discharge standards (<0.5 mg/L), while also demonstrating some effectiveness in removing Sb [19]. This method has the advantages of good selectivity and high removal efficiency.

Although each of these methods has its own characteristics, some may face issues such as high iron ion content in the effluent, the generation of high-salinity wastewater, poor economic efficiency, or secondary pollution [15,20]. Moreover, most methods do not focus on the recovery aspect during the removal of antimony.

Electrochemical methods, known for their environmental friendliness and efficiency, are widely applied in the treatment of high-concentration heavy metal wastewater. The primary electrochemical technique used for treating antimony-containing wastewater is electrocoagulation, which can reduce Sb(III) and Sb(V) concentrations to 5.0 and 28.1 µg/L, respectively, after 30 minutes of treatment [21]. Further research has optimized the electrocoagulation process, showing that with a reaction time of 20 minutes and a current density of 13.89 A/m², iron-based electrocoagulation achieves a higher Sb removal efficiency, approximately 14.02% greater than that of aluminum-based electrocoagulation [22]. However, electrocoagulation may have issues such as secondary pollution and recycling. Another electrochemical treatment method is electrochemical hydride generation, which is currently mainly used for sample measurement technology, with some applications in wastewater treatment. Bejan et al. used a tandem electrolytic device to treat As(III) (100 mg/L) of the same group as Sb under acidic and alkaline conditions using a glass tube carbon electrode, reducing the As concentration to 20 µg/L [23]. Chen et al. realized the removal of simulated Sb-containing wastewater (5 mg/L) under acidic conditions using a cation membrane electrolytic device while achieving Sb recovery with a tail-end heating device [24]. Notably, in electrolysis systems, the membrane serves as a critical component of the apparatus, with its performance potentially having a direct impact on removal efficiency. The proton exchange membrane, which is widely used in the chlor-alkali industry and fuel cells, exhibits excellent chemical stability and electrochemical performance. Its strong proton conductivity has the potential to significantly influence the transformation behavior of antimony species, thereby affecting the overall removal efficiency. Additionally, various operational factors in such systems, including temperature and current density, can potentially impact the removal efficiency in electrochemical processes. Therefore, this study explores the advantages of a novel Nafion membrane electrolytic device for Sb removal and further analyzes the effects of temperature, stirring method, current intensity, and electrode surface area. The aim is to identify the influence patterns of these conditions while enhancing the removal efficiency and reducing the need for pH control, ultimately providing optimization directions for future applications in treating actual mine wastewater.

## 2. Materials and methods

### 2.1 Experimental devices and materials

As shown in Fig 1, the apparatus consists of two functional units: a removal unit and a recovery unit. The removal unit is configured as an electrolytic cell with a total volume of 100 mL, where each compartment has a volume of 50 mL. The electrolyte containment volume is 45 mL. The power supply was an adjustable DC regulated power source (TPR3010S, ATTEN Instruments Corp., Guangdong, China). The assembly method of the Nafion membrane (N117, DuPont) used in this experiment (Fig 2a) was a "sandwich" structure, with the middle layer being the Nafion membrane and the sides being platinum electrodes. The platinum electrode was a plate electrode (5 × 5 cm), with dimensions slightly smaller than the membrane area. Square openings are made in the center of the electrode according to area requirements (Fig 3). The electrode is fully immersed in the liquid. The traditional electrolytic cell used in the experiment (Fig 2b) features identical dimensions to the proton exchange membrane (Nafion membrane) electrolytic cell (Fig 2a). The traditional cell employed platinum electrodes, utilized a cation exchange membrane as the separator, and adopted $K_2SO_4$ (0.4 mol/L) as the electrolyte. The recovery unit consists of a quartz tube and high-temperature heating wires. The Sb-containing wastewater (Sb(III)) to be treated was placed in the cathode chamber, which was then sealed and connected to a heating recovery unit to prevent the escape of gas. Traditional electrochemical methods (e.g., electrocoagulation) for treating metal-containing wastewater primarily operate at the anode. Flocs (such as $Fe(OH)_3$) formed through anode oxidation adsorb target metals or induce coprecipitation for removal. Heavy metals in the resulting precipitates remain difficult to recover. This system fundamentally differs from traditional electrochemical approaches in Sb removal/recovery:

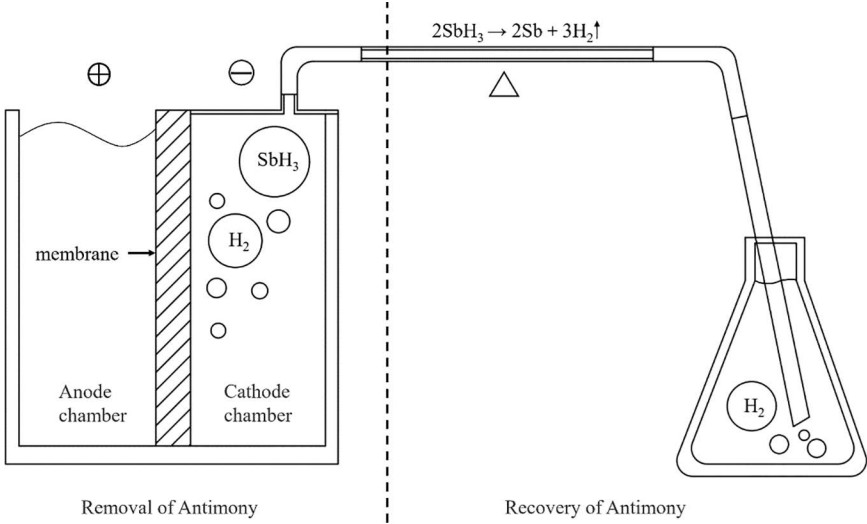

**Fig 1. Removal and recovery apparatus for antimony-containing wastewater.**

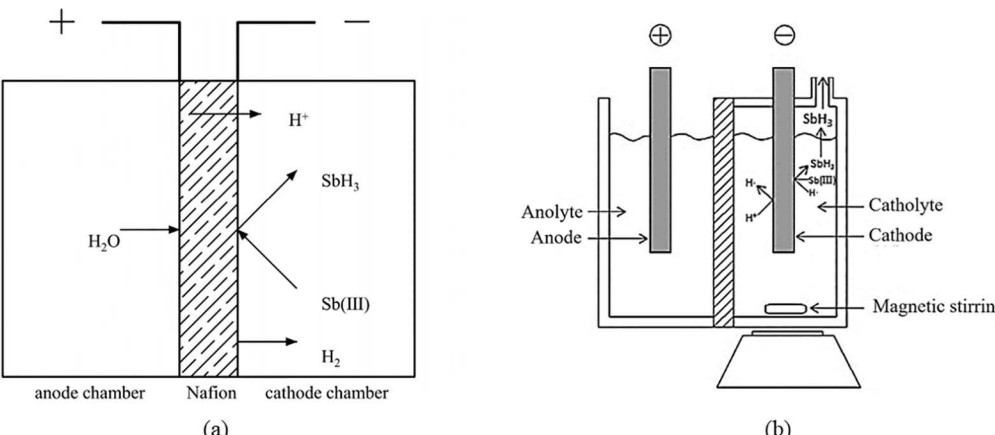

**Fig 2. Schematic diagram of antimony-containing wastewater removal unit.** (a) Nafion membrane electrolytic cell; (b) traditional membrane electrolytic cell [24].

(1) Cathodic removal: Sb combines with H under electric fields to form volatile stibine gas ($SbH_3$) via the half-reaction: $Sb^{3+}(aq) + 3H^+ + 6e^- \rightarrow SbH_3(g)$ [25]. This process eliminates secondary contamination risks from sludge and prevents cathode material consumption.

(2) Resource recovery: Collected $SbH_3$ undergoes thermal decomposition: $2SbH_3(g) \rightarrow 2Sb(s) + 3H_2(g)$, enabling direct Sb reclamation.

All water used in the experiments was ultrapure (resistivity: 18.2 MΩ·cm), produced by a Milli-Q system (Advantage A10, Merck). The preparation of synthetic wastewater was based on studies of Sb concentrations in contaminated water bodies from China's largest antimony mine, the Xikuangshan Mine in Hunan, the typical Banpo Antimony Mine in Guizhou, and antimony mines in Slovakia. Specifically, Sb concentrations in the leachate from the tailings dam and the smelter

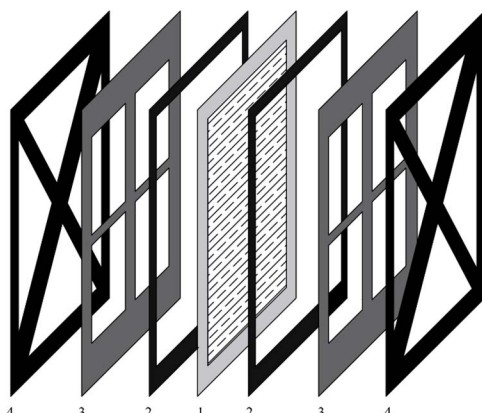

**Fig 3. Assembly diagram of Nafion membrane.** 1-Nafion membrane; 2 and 4-Fixed plates; 3- Platinum electrode.

wastewater at Xikuangshan were 6.9 and 14.5 mg/L, respectively [26]. The Sb concentrations in the tailings leachate and mine pit water at Banpo Antimony Mine were 7.9 and 1.3 mg/L, respectively [8]. In Slovakia, the highest Sb concentration in various contaminated water bodies from five abandoned antimony mining areas was 9.3 mg/L [27]. Therefore, a suitable value was selected for this study, using potassium antimony tartrate ($KSbC_4H_4O_7$, purity≥99%, Fluka, Germany) to prepare the synthetic Sb-containing wastewater with an Sb(III) concentration of 5 mg/L. The buffer solution used in the pH control experiment was pH = 7, prepared by dissolving 13.6 g of $KH_2PO_4$ and 22.8 g of $K_2HPO_4$ in water and diluting to 100 mL. All chemical reagents used in the experiments were of analytical grade.

## 2.2 Experimental procedures

Each experimental run lasted for 2 hours, with all tests conducted in triplicate. The study systematically investigated the effects of membrane material, temperature, stirring method, current density, and electrode area on Sb removal efficiency.

(1) To investigate the advantages and disadvantages of the Nafion membrane electrolytic device in Sb wastewater treatment, comparative experiments were conducted with a traditional ion-exchange membrane electrolytic cell. The traditional electrolytic cell was used to perform controlled and uncontrolled pH experiments. The Nafion membrane electrolytic cell was used only to conduct the uncontrolled pH experiment. In the controlled pH experiment, the pH was adjusted to 7 using a buffer solution. Both types of electrolytic cell experiments were conducted at the same current intensity (0.5 A) and electrode area. After analyzing the advantages of the Nafion membrane electrolysis device for Sb removal, this device was further used to investigate the effects of other factors on the Sb removal efficiency.

(2) The experiment used a circulating water bath method for temperature control. The electrolytic cell was placed in the self-made circulating water cooling temperature control device, and the external circulating water temperature was set. The experiment began when the temperature of the solution in the electrolytic cell was consistent with the external circulating water temperature. Five circulating water temperature levels were set: 2, 15, 30, 40, and 50 °C.

(3) Three groups of stirring method experiments were set: (1) no stirring during electrolysis; (2) magnetic stirring: with a magnetic stirrer placed in the cathode chamber solution, the stirring speed was maintained at 800 rpm; and (3) ultra-sonic stirring: with an ultrasonic probe placed in the cathode chamber solution, the ultrasonic frequency was set to 20 kHz. To ensure that the temperature inside the electrolytic cell did not continuously rise due to ultrasonic heating, the experiment utilized 8-second pulse ultrasonication cycles, with each ultrasonication period lasting 5 minutes and an interval of 5 minutes between cycles.

(4) To optimize the current density and current intensity in the Sb treatment process, an orthogonal experimental design was employed, and the results were subsequently fitted. The current density was calculated by dividing the current intensity by the electrode area. The platinum electrodes were made into sheets according to the shape of the electrolytic cell and perforated according to the required experimental area. We performed sampling and calculated removal rates after 2 hours of the experiment. In terms of condition settings, a total of 15 groups of experiments were set. The experimental conditions are shown in Table 1.

## 2.3 Analysis

The pH of the samples was measured using a Denver UB-7 pH meter (TOA-DKK, Japan). The concentration of total antimony (Sb(T)) was analyzed by the method in Zhang et al. [28]. Ascorbic acid (10 g/L) and thiourea (10 g/L) were mixed with the collected samples to reduce pentavalent Sb to trivalent species and to mask the interference of metals, and then the concentration of Sb(T) was determined by hydride generation-atomic fluorescence spectrometry (HG-AFS) (AFS-2202E, Haiguang Instruments Corp., Beijing, China). For the analysis of Sb(III), citric acid (1 g/L) was added to mask the Sb(V) signal, and the concentration of Sb(III) was then determined. The Sb(V) concentration was calculated as the difference between the Sb(T) and Sb(III) concentrations [29]. The detection limits for Sb(T) and Sb(III) in AFS measurements were 0.004 and 0.05 µg/L, respectively, with relative standard deviations of 0.5% and 0.6%. To ensure no gas loss, the detection of recovered antimony was conducted separately. Sampling was not performed at time intervals during the experiment. The temperature was maintained at ≥ 200 °C, and the electrolysis duration was set to 2 hours. After the recovery experiment, the electrodes, exchange membrane, and quartz tube were removed and soaked in concentrated nitric acid for 12 hours. The method for antimony detection was the same as described previously.

## 3. Results and discussion

### 3.1 Comparison of Sb removal efficiency between nafion and traditional membrane electrolytic cells

The experiment compared changes in pH and Sb concentration under three different conditions. As shown in Fig 4a, with a traditional cation exchange membrane under uncontrolled pH conditions, the pH of the catholyte rapidly increased to a strong alkaline state (pH = 12.34) within a short period (10 minutes). This is due to the consumption of a large number of hydrogen ions during the electrolysis process. When a buffer solution was used to control the pH, the pH of the catholyte remained near neutral for 80 minutes but then increased significantly. Notably, when Nafion was used as the membrane, after 2 hours of electrolysis, the pH of the catholyte changed only from 5.73 to 6.71, indicating minimal pH change. This was because the water electrolysis process occurs at the anode surface of Nafion (reaction: $H_2O \rightarrow 2H^+ + 2e^- + 1/2O_2$), and Nafion membranes have strong proton conductivity. Therefore, the produced hydrogen ions directly reach the cathode surface where they either produce hydrogen gas or combine with electrons delivered to the cathode to form H·, which then further reacts with other substances. The electrolysis process essentially does not disrupt the balance of $H^+$ and $OH^-$ in the solutions at both electrodes [30,31].

Regarding Sb removal, after 120 minutes of electrolysis, the residual Sb concentrations under the three conditions were 3.04 ± 0.22 (uncontrolled pH), 2.34 ± 0.25 (controlled pH), and 1.41 ± 0.11 mg/L (Nafion), and the removal rates were

**Table 1. Experimental group design.**

| Group number | 1 | 2 | 3 | 4 | 5 | 6 | 7 | 8 | 9 | 10 | 11 | 12 | 13 | 14 | 15 |
|---|---|---|---|---|---|---|---|---|---|---|---|---|---|---|---|
| Current intensity(A) | 0.1 | 0.1 | 0.1 | 0.1 | 0.1 | 0.3 | 0.3 | 0.3 | 0.3 | 0.3 | 0.5 | 0.5 | 0.5 | 0.5 | 0.5 |
| Electrode area(cm²) | 5 | 10 | 16 | 20 | 25 | 5 | 10 | 16 | 20 | 25 | 5 | 10 | 16 | 20 | 25 |

**Fig 4. Variation in cathode pH (a) and Sb concentration (b) over time under different membrane conditions.**

39.2±4.4, 53.2±5.0, and 71.8±2.2%, respectively. Without electrolysis, the Sb concentration in the solution showed no significant changes, a phenomenon consistently observed in subsequent experiments (Fig 4b). Experimental results indicate that antimony primarily exists as Sb(III) during electrolysis (Fig 5), with no significant oxidation observed. The removal mechanism involves active hydrogen atoms (H•) generated electrolytically reacting with Sb(III) to form $SbH_3$. Due to its low aqueous solubility, $SbH_3$ escaped the system with hydrogen gas. Recovery experiments revealed an antimony

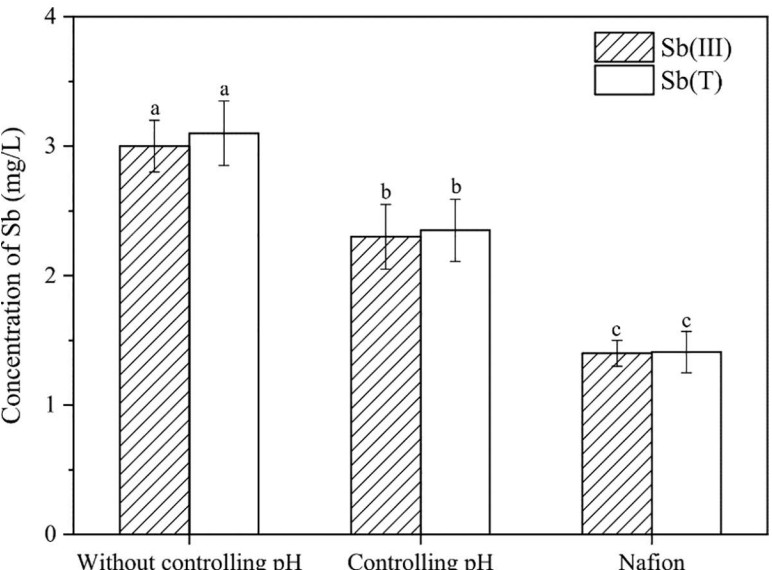

**Fig 5. Effects on Sb oxidation states after treatment under different membrane conditions (after 120 minutes).** The removal efficiency differences among various membrane conditions were tested by one-way analysis of variance (ANOVA), indicated by different letters on the top of column (Duncan; $p < 0.05$).

mirror phenomenon on the quartz tube (S1 Fig), and subsequent hot nitric acid treatment confirmed that up to 71.8±3.6% of antimony accumulated in the quartz tube (S2 Fig), confirming $SbH_3$ generation as the primary Sb removal pathway. The current recovery rate of 67.3±2.9–71.8±3.6% suggests that parameters such as gas flow rate, decomposition efficiency, and recovery temperature require further optimization—a key focus for future research. For future large-scale production, direct harvesting of thermally deposited antimony metal could serve as a viable operational strategy. As shown in Fig 4b, compared to the traditional electrolytic cell, the residual Sb concentration was always lowest when using the Nafion membrane. Based on the trend of pH changes over time (Fig 4a), it is speculated that the neutralization of a large amount of H in the alkaline environment slowed down the formation of $SbH_3$. Furthermore, it was found that as the pH increases, the adsorption of $H_2O$ on the electrode surface decreases, leading to a reduced hydrogen evolution efficiency, and the apparent adsorption strength of H increases, which is not conducive to further participation in the formation of $SbH_3$ [32,33]. In the later stage of the controlled pH experiment (time > 80 minutes), the significant increase in pH and the decrease in Sb removal rate (Fig 4b) also confirm this inference. In summary, the proton-sieving selectivity and high proton conductivity of the Nafion membrane collectively contribute to electrolyte pH stabilization, thereby enhancing $SbH_3$ generation efficiency within the weakly acidic range (pH 5.73–6.71) and consequently improving antimony removal efficacy. This indicates that the operation process of the Nafion membrane electrolytic cell in antimony treatment is simpler and more conducive to widespread application.

## 3.2 Effect of temperature on Sb removal

Since the electrolytic process induces continuous temperature variations in the reaction solution, which may adversely affect chemical reaction kinetics or membrane conductivity, a temperature-controlled analytical approach was implemented to systematically investigate the variation patterns of contaminant removal efficiency under differing operational conditions. Experiments were conducted using the Nafion membrane electrolytic cell at five different circulating water temperatures (2, 15, 30, 40, and 50 °C). After approximately 30 minutes, the temperature of the cathode chamber solution equilibrated with the circulating water temperature and stabilized under controlled experimental conditions.

As shown in Fig 7, after 120 minutes, the Sb removal rates were 74.2±4.4% (2 °C), 72.2±3.0% (15 °C), 87.3±2.6% (30 °C), 79.8±5.0% (40 °C), and 75.6±4.6% (50 °C). Overall, the removal efficiency increased with temperature up to a certain point before decreasing. Specifically, at 30 °C, the residual Sb concentrations were consistently lower compared to lower temperatures (2 and 15 °C), with removal rates 10–15% higher (Fig 6). This phenomenon may be due to increased molecular activation at higher temperatures, which facilitated the conversion reaction of Sb(III) to $SbH_3$. Other studies have found that an increased operating temperature leads to a reduced Open-Circuit Voltage (OCV) and reduced kinetic losses due to improved reaction and transport kinetics, resulting in lower energy demands for electrolysis [34,35]. Additionally, higher temperatures reduced the solubility of slightly soluble $SbH_3$, facilitating its escape. These factors together contribute to the increased removal rate at higher temperatures. As the temperature increased beyond 30°C, residual Sb concentrations during the initial reaction phase (0–25 min) at 40°C and 50°C remained comparable to those observed at 30°C, confirming the temperature-dependent synergistic effects on removal efficiency. However, beyond the 25 min, both 40°C and 50°C systems exhibited consistently higher residual Sb concentrations than the 30°C system (Fig 6). This may be related to the decomposition of $SbH_3$, which has thermal stability greater than $BiH_3$ but less than $AsH_3$, and can decompose vigorously at 200 °C. Studies have shown that $SbH_3$ already has a certain decomposition rate at 25 °C, and hydrogen gas content had little effect on the reaction rate [36,37]. It can be inferred that further increasing the temperature may promote the decomposition of $SbH_3$ that has not yet been released (with the possible reaction being: $SbH_3 + 3H_2O \rightarrow Sb(OH)_3 + 3H_2$), causing some Sb to re-enter the solution. Therefore, within the operational temperature range below 30°C, elevating system temperature enhances antimony removal efficiency. However, when exceeding this threshold, the negative impact of $SbH_3$ re-dissolution progressively dominates, leading to reduced removal rates.

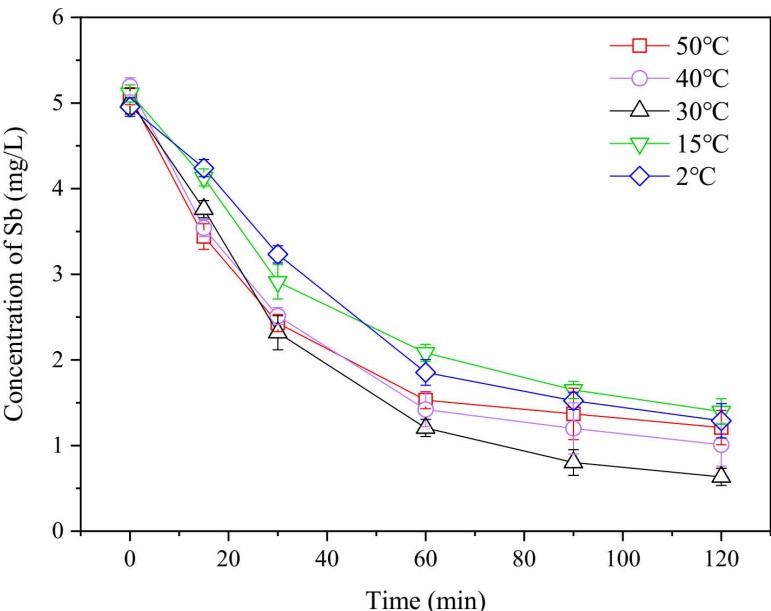

**Fig 6. Variation in Sb concentration over time under different temperature controls.**

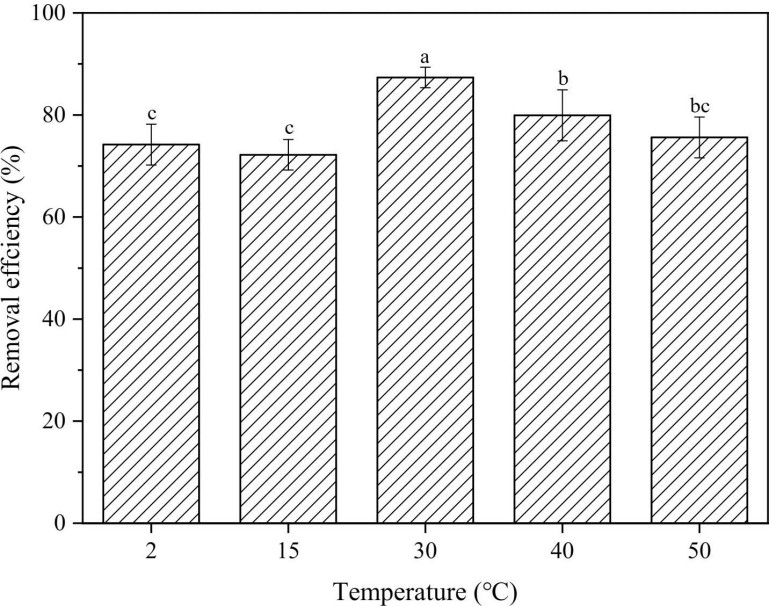

**Fig 7. Removal efficiency of Sb at different temperatures (after 120 minutes).** The removal efficiency differences among the five temperatures were tested by one-way analysis of variance (ANOVA), indicated by different letters on the top of column (Duncan; $p < 0.05$).

### 3.3 Effect of stirring method on Sb removal

To enhance ion migration within the solution, mitigate concentration polarization near electrode surfaces, and sustain reactive species diffusion toward the electrodes for improved process efficiency, a comparative study was conducted to

evaluate the effects of distinct stirring methods (magnetic stirring and ultrasonic stirring) on Sb removal efficacy. To minimize temperature variations among the three experimental groups, ultrasonic stirring was applied intermittently. Temperature changes under different stirring conditions are shown in Fig 8a. During ultrasonic stirring, the solution temperature was approximately 10 °C higher than in other groups, but it returned to nearly the same level after ultrasonic stirring was stopped.

The residual Sb concentrations under the three stirring conditions were 1.40±0.15 mg/L (no stirring), 1.21±0.20 (magnetic stirring), and 1.32±0.11 mg/L (ultrasonic stirring), with corresponding removal rates of 71.9±3.0, 75.8±4.0, and 73.6±2.2%. Ultrasonic stirring showed a significantly faster removal rate in the early stages compared to the other two methods (Fig 8b). Specifically, after 30 minutes, ultrasonic stirring reduced the Sb concentration to 2.21±0.12 mg/L, which is about 60% more efficient than the other two groups (no stirring: 3.30±0.23 mg/L; magnetic stirring: 3.22±0.10 mg/L). During experimentation, bubble displacement on electrode surfaces was markedly observed in the ultrasonic stirring group. This suggests ultrasound cavitation achieves dual effects: (1)Continuous electrode surface cleaning (maintaining activity), (2)Increased effective electrode area. Simultaneously, this process reduced the residence time of $SbH_3$ in solution. Such reduction helps prevent $SbH_3$ re-dissolution, thereby improving removal efficiency. Additionally, when the solution pH was near neutral, Sb(III) was likely present in forms such as $SbO^+$ and $SbO_2^-$. The high-temperature and high-pressure micro-environment created by cavitation might weaken the stability of the Sb-O bonds in Sb(III), making it easier to react with the hydrogen radicals (H•) generated during electrolysis to form $SbH_3$. The resulting $SbH_3$, which is slightly soluble in water, was carried away by bubbles and was less affected [9,38]. Furthermore, cavitation also promotes continuous movement of the solution near the electrode, reducing the concentration polarization and accelerating chemical reactions to some extent.

However, in the later stages of the experiment (90–120 minutes), the removal effect of ultrasonic stirring (from 1.40 mg/L to 1.32 mg/L, a reduction of 0.08 mg/L) was lower than that of magnetic stirring (from 1.43 mg/L to 1.21 mg/L, a reduction of 0.22 mg/L) (Fig 8b). This could be due to the lower reaction rate at lower concentrations and the increasingly significant negative effects of $SbH_3$ re-dissolution caused by the heat generation effect of ultrasound. Additionally, the

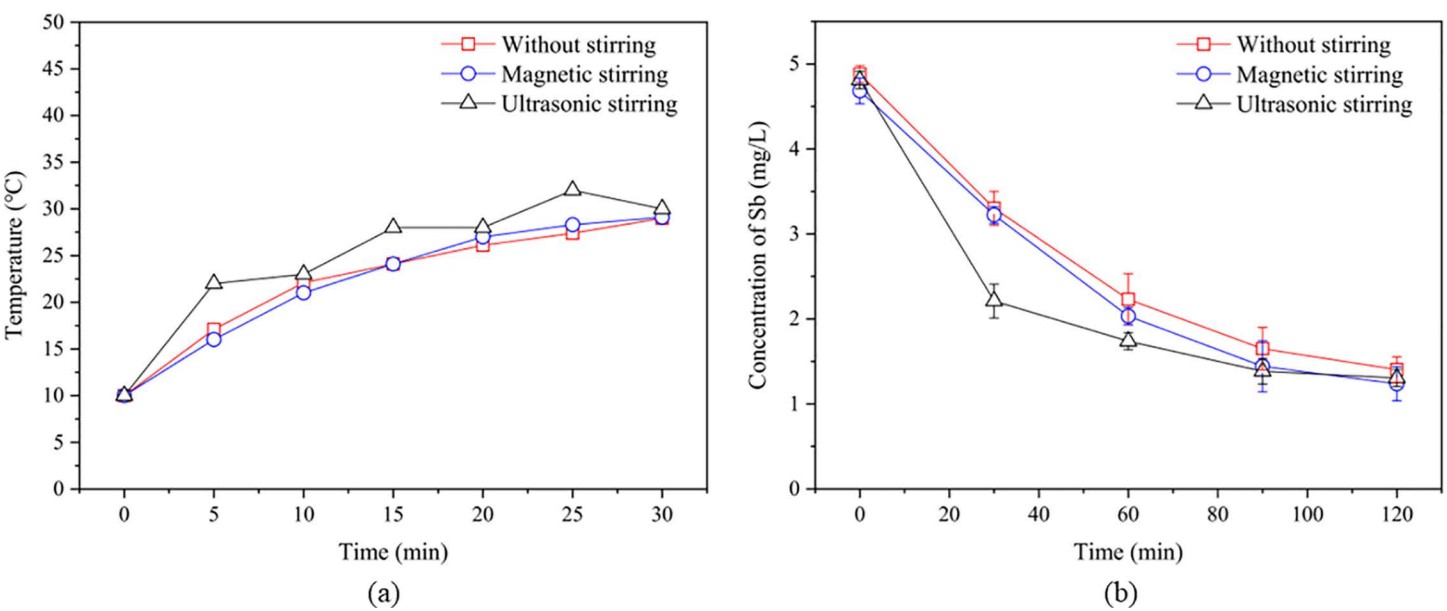

**Fig 8. Variation in temperature (a) and Sb concentration (b) over time under different stirring methods.**

ultrasonic process may accumulate trace amounts of strong oxidizers ($H_2O \rightarrow H\bullet + HO\bullet$, $H\bullet + O_2 \rightarrow HO_2\bullet$, $H_2O\bullet + H\bullet \rightarrow H_2O_2$, $H_2O\bullet + H_2O \rightarrow H_2O_2 + O_2$), which could hinder the further reduction of Sb(III) and affect the removal efficiency [39,40]. Therefore, ultrasonic stirring is suitable for accelerating removal rates at higher Sb concentrations, while electromagnetic stirring is more appropriate for long-term treatment at lower concentrations.

### 3.4 Effect of current intensity and electrode area on Sb removal efficiency

The experiment used an orthogonal experimental method to investigate the effects of current intensity and electrode area on removal efficiency. The results were plotted using the adjacent averaging method to generate a surface fitting graph (Fig 9). A second-order polynomial regression equation for the relationship between current intensity, electrode area, and Sb removal efficiency was fitted using Design-Expert 13:

$$Y = 54.28 + 37.59A + 1.3B + 2AB - 87.73A^2 - 0.05B^2$$

where Y is the removal efficiency (%), A is the current intensity (A), and B is the electrode area (cm²). In this experiment, the p-values for the linear terms (A, B), interaction term (AB), and quadratic terms (A², B²) were all less than 0.01 (S1 Table), indicating that the effects of current intensity and electrode area on removal efficiency were highly significant. The model p-value was less than 0.01, indicating that the model was highly significant. The difference between the predicted

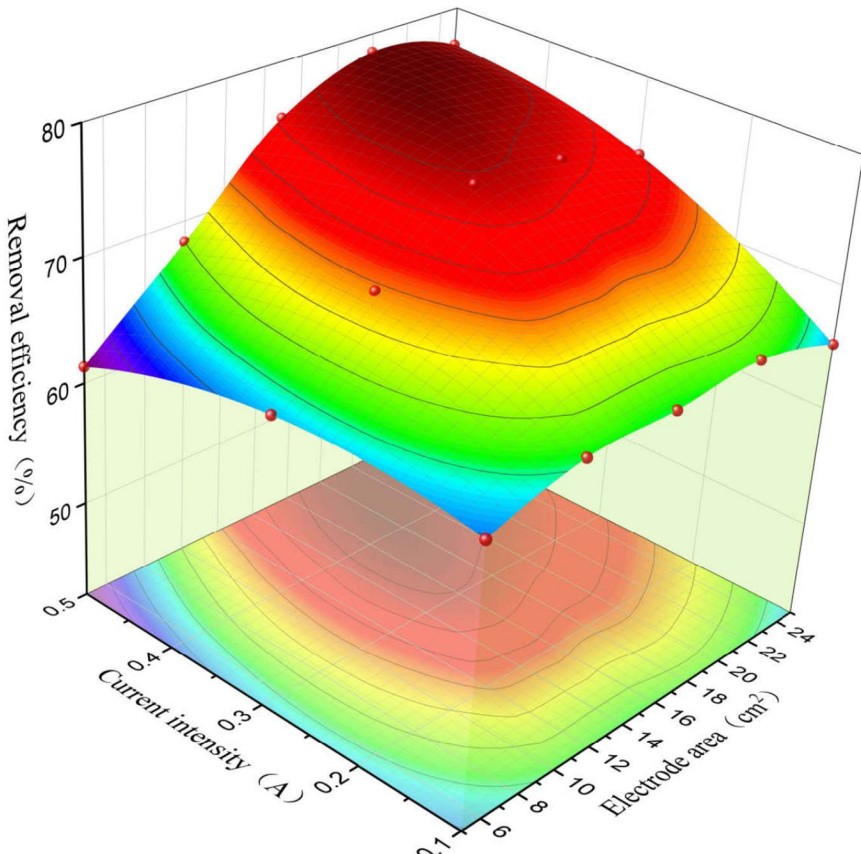

**Fig 9. Influence of current intensity and electrode area on Sb removal efficiency.**

R² (0.88) and adjusted R² (0.94) was less than 0.2, the coefficient of variation (C.V.%) was 1.76 (less than 10), and the signal-to-noise ratio was 21.15 (greater than 4) (S2 Table), suggesting that the regression model could adequately describe the experimental process and that the fitted regression equation had good adaptability. The optimal reaction conditions predicted under the experimental conditions (current intensity < 0.5 A, electrode area < 25 cm²) were a current intensity of 0.47 A and an electrode area of 23 cm², with a predicted optimal Sb removal efficiency of 78.1%. Verification of this optimized experiment yielded a removal efficiency of 77.6 ± 4.6%, which is close to the predicted value, indicating that the model had a certain degree of reliability and can effectively predict the impact of current intensity and electrode area on Sb removal from water.

As shown in Fig 9, when the current intensity is low (0.1 A), changes in electrode area have a minimal effect on removal efficiency (64.7–67.5%). However, as the current intensity increases (0.3 and 0.5 A), the removal efficiency initially increases with increasing electrode area but then decreases. This phenomenon may be due to the low reaction rate at low current intensities at which the electrode area has a negligible effect on the reaction. At higher current intensities, the reaction rate increases, and the larger electrode area significantly enhances the contact probability between H• on the electrode and Sb in the liquid, thus effectively accelerating the formation of $SbH_3$. However, further increasing the electrode area will reduce the current density, and the hydrogen evolution overpotential will become increasingly influenced by the current density, showing a positive correlation, which further increases hydrogen production [41]. Notably, the production of hydrogen and the formation of $SbH_3$ are competitive processes, leading to a decrease in the yield of $SbH_3$. This trend can also be clearly seen from the relationship between current density and removal efficiency shown in Fig 10, where both excessively high and low current densities are unfavorable for improving removal efficiency. The higher removal efficiency (78.5 ± 4.6%) is achieved when the current density is around 25 mA/cm² (current intensity of 0.5 A, electrode area of 20 cm²). Along the electrode area axis, in lower electrode areas (≤16 cm²), the removal efficiency increases with current intensity up to a point and then decreases. However, when the electrode area is increased (20 and 25 cm²),

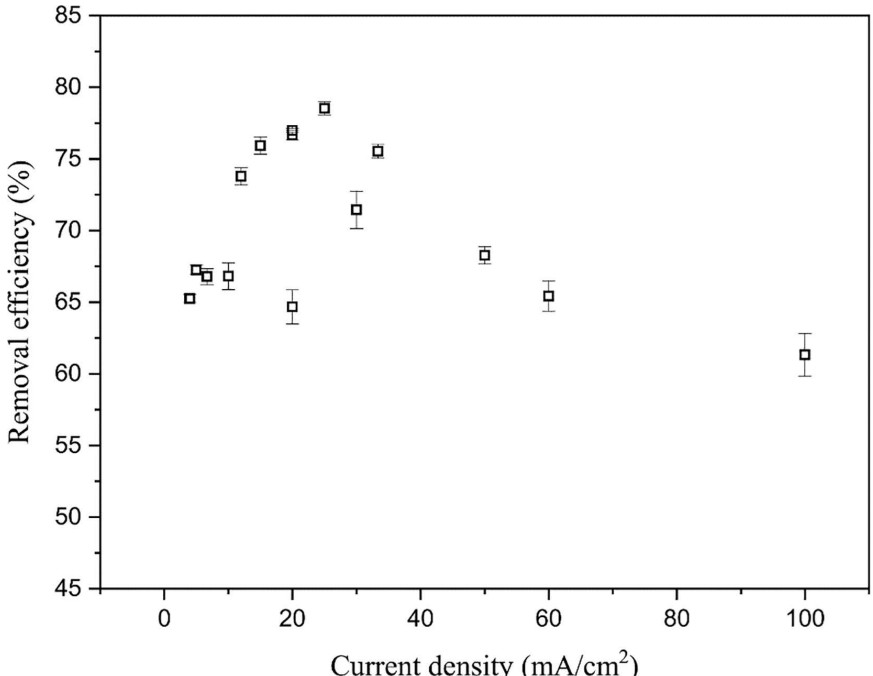

**Fig 10. Influence of current density on Sb removal efficiency.**

the removal efficiency increases with current intensity (Fig 9). This may be related to the gas-blocking effect. In low electrode areas, the current density directly determines the volume increase of gas bubbles per unit time. Especially at high current density, the bubble diameter increases rapidly, and the bubbles detach with larger diameters [42,43]. Therefore, slightly increasing the current intensity in a low electrode area may accelerate the reaction rate. However, further increasing the current intensity will increase the current density, potentially triggering premature gas-blocking effects. These effects significantly hinder the interaction between Sb and the H· on the electrode surface, thereby reducing the removal efficiency. Conversely, this phenomenon is less likely to occur under high electrode area conditions.

In terms of current efficiency, the calculation is as follows:

$$CE(\text{Current Efficiency}) = \frac{\text{mol}_{product} \times nF}{It}$$

where $\text{mol}_{product}$ is the amount of target product $SbH_3$ (converted from the amount of Sb removed), n is the number of transferred electrons, F is Faraday's constant (96485.34 C/mol), I is the current value set in the experiment, and t is the reaction time. As shown in Fig 11, the current efficiency is primarily influenced by the current intensity, with minimal impact from the electrode area. As the current intensity increases (from 0.1 to 0.5 A), the current efficiency steadily decreases (from $0.218 \pm 0.004$ to $0.048 \pm 0.005$). This decline may be attributed to the massive hydrogen evolution triggered by increased current intensity, while the increment of $SbH_3$ remained negligible. Under the optimal removal current density (0.5 A/20 cm², 45 mL) with 2h treatment, the system exhibited a high unit energy consumption of approximately 104.4 kWh/m³. This high

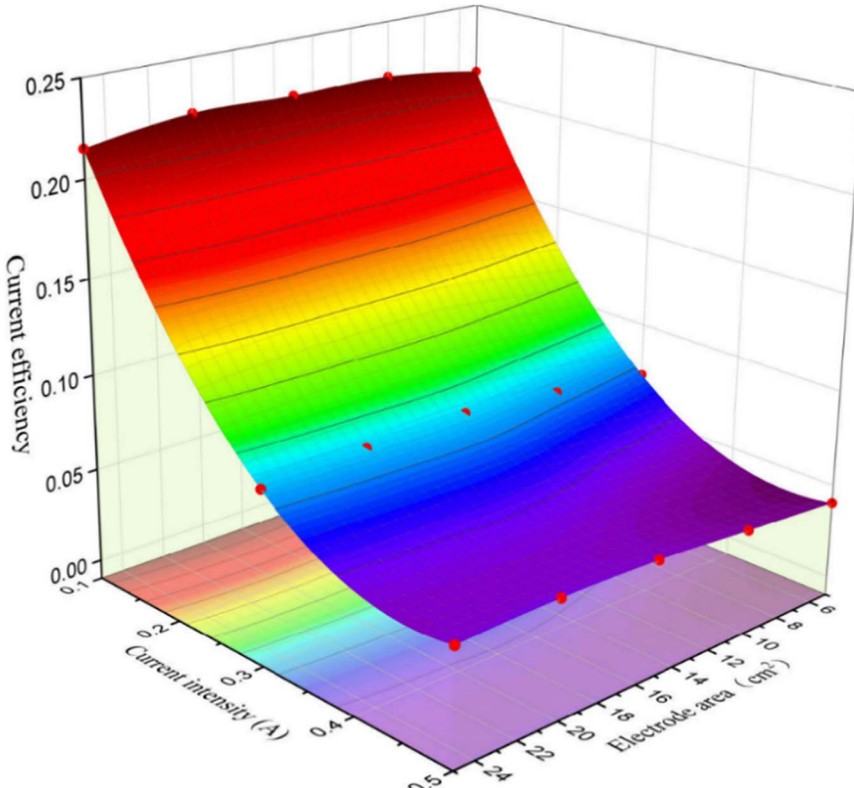

**Fig 11. Influence of current intensity and electrode area on current efficiency.**

energy consumption primarily resulted from extremely low current efficiency (merely 0.048) due to competing hydrogen evolution reactions. As previously discussed, although a higher current intensity can achieve higher removal rates, the majority of the consumed electrical power is expended on hydrogen generation. Therefore, when employing the electrochemical hydride generation method for Sb treatment, it is imperative to explore strategies to reduce hydrogen production to enhance energy efficiency and improve Sb removal rates.

**3.5  Implications for the application in actual mine wastewater treatment**

The electrochemical hydride generation method can be applied to treat Sb-containing wastewater, with a heating device used for the recovery of metallic antimony. This study explores the effects of factors such as membrane selection, temperature, stirring methods, and current intensity on Sb removal, providing guidance for optimizing this method (S3 Table).

To preliminarily assess the feasibility of this technology for treating actual wastewater, leachate from the tailings of the Banpo Antimony Mine was collected for experimentation, with the experimental conditions set as described in section 3.1. The leachate collection pit had been continuously treated with quicklime for alkalization to precipitate common heavy metals, resulting in a slightly alkaline pH (7.76). The leachate primarily contains Sb and Fe. A comparative analysis of initial pH, ionic composition, and removal rates relative to synthetic wastewater is provided in S4 Table. The results, shown in S3 Fig, indicated that Sb removal was still achieved (removal rate of 48.7±5.4%), although the removal rate was lower compared to that for synthetic wastewater (71.8±2.2%, as presented in section 3.1). Although the leachate was collected from the bottom water layer, the alkaline conditions promoted metal oxidation, resulting in partial existence as high-valence states (Sb(V) accounting for >60% of total Sb) (S4 Table). During the electrolysis process, the reduction of these high-valent metals likely decreased the overall reaction efficiency. Iron deposition on the cathode passivated the electrode and may even block proton channels, which could be one of the critical factors affecting removal efficiency. Furthermore, the alkaline conditions (7.76) and relatively low Sb concentration may have reduced the generation efficiency of $SbH_3$,. Therefore, when applying this method to treat antimony-containing wastewater, careful consideration should be given to the effects of pH value, antimony speciation, as well as interference from other metal ions or anions [22,23]. Practical applications may benefit from the use of appropriate pre-treatment devices or multi-stage treatment processes to reduce the concentration of impurity metals and lower the pH. In terms of safety, $SbH_3$ is a highly toxic and flammable gas that poses risks of acute poisoning upon inhalation. Therefore, gas detectors and positive pressure respirators must be equipped for protection during large-scale operations. In addition, the economic aspects are crucial in the application of this technology, including the recovery value of metallic antimony, the potential use of hydrogen generated during electrolysis, and the cost reduction achieved by not requiring additional reagents. Following the validation of the Nafion membrane electrolysis system's efficacy in actual wastewater treatment, future research will prioritize membrane modifications and the development of hybrid ultrasonic-electrochemical reactors.

## 4.  Conclusions

(1)  This study advances electrochemical remediation by demonstrating Electrochemical Hydride Generation Technology as a dual-functional solution for Sb removal and recovery, contributing to Sb resource recovery in wastewater treatment. Compared to traditional membrane electrolytic devices (which achieve a removal rate of 53.2±5.0%), Nafion membrane electrolytic devices do not require pH control and achieved a higher removal efficiency of 71.8±2.2%.

(2)  Increasing the reaction temperature can accelerate the generation and release of $SbH_3$, which can improve the removal rate to some extent (87.3±2.6% at 30 °C). However, excessively high temperatures (above 30 °C) may lead to the re-dissolution of $SbH_3$, thereby reducing the removal rate.

(3)  When treating wastewater containing high concentrations of Sb (5 mg/L), ultrasonic stirring can significantly accelerate the generation and release of $SbH_3$. However, as concentrations progressively decrease, the adverse effects of

ultrasonic heating and the contribution of trace oxidizers become more pronounced. Ultimately, this results in no significant enhancement in removal efficiency compared with magnetic stirring.

(4) Current intensity and electrode area significantly affect the removal rate. The higher removal rate (78.5±4.6%) can be achieved with a current intensity of 0.5 A and an electrode area of 20 cm². Although the removal rate of Sb is higher at high current intensities, the device's current efficiency was lower, indicating that a substantial amount of electricity is used for hydrogen production. To improve removal efficiency and reduce energy consumption, further studies exploring methods to reduce hydrogen production are required.

## Supporting information

**S1 Fig. The silver mirror phenomenon produced in the recovery device.**
(TIF)

**S2 Fig. The recovery rate of Sb in Nafion membrane electrolytic device (a) and traditional membrane electrolytic device (b).**
(TIF)

**S3 Fig. Variation of Sb concentration and cathode pH with time in the treatment of real wastewater.**
(TIF)

**S1 Table. ANOVA for quadratic model.**
(DOCX)

**S2 Table. Fit statistics.**
(DOCX)

**S3 Table. The impact of optimized conditions on Sb removal.**
(DOCX)

**S4 Table. Compositional comparison of actual wastewater versus synthetic wastewater.**
(DOCX)

## Author contributions

**Conceptualization:** Jingjing Chen, Guoping Zhang.

**Formal analysis:** Jingjing Chen.

**Funding acquisition:** Jingjing Chen, Yan Xiong.

**Supervision:** Yi Bai, Yan Xiong.

**Visualization:** Jingjing Chen.

**Writing – original draft:** Jingjing Chen.

**Writing – review & editing:** Xing Liu.

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
