## [Decision Letter · Decision Letter 0]

17 Jun 2025

Dear Dr. Liu,

Thank you for submitting your manuscript to PLOS ONE. After careful consideration, we feel that it has merit but does not fully meet PLOS ONE’s publication criteria as it currently stands. Therefore, we invite you to submit a revised version of the manuscript that addresses the points raised during the review process.

We look forward to receiving your revised manuscript.

Kind regards,

Hailing Ma

Academic Editor

PLOS ONE

Journal Requirements: 

Additional Editor Comments:

Reviewer Comments – Major Revision (Total: 50)

Abstract Structure & Scientific Clarity (1–10)

Lack of clarity in the problem statement: The abstract does not clearly state why antimony removal is an urgent or novel topic. Please briefly introduce the environmental risk.

Improve technical flow: The transition between electrolysis, SbH₃ formation, and Sb recovery feels abrupt—provide a clearer cause-effect structure.

Quantify the initial concentration ranges: State the initial Sb concentrations used in the experiments for context.

Avoid passive phrasing: Revise passive structures like “was employed” to active voice where possible.

Explain recovery vs. removal distinction: Is Sb merely removed or also recovered in solid form? Clarify in abstract.

Clarify mechanisms: State more explicitly how SbH₃ formation and decomposition enable Sb separation and recovery.

Reword ambiguous phrasing: "Overflowed from the solution" is vague; suggest "was released as gas" or similar.

Include brief methodological note: Mention reactor configuration or stirring methods used, even if in brief.

Summarize results more clearly: Mention the optimum conditions in one concise summary sentence.

Add significance: Conclude abstract with a statement on the broader implications for wastewater treatment.

Technical Depth and Experimental Design (11–20)

Compare to existing methods: Clarify how this method compares to conventional chemical precipitation or adsorption approaches.

Explain electrode selection: What electrode materials were used? Were they inert or catalytic?

Include membrane specification: Provide model or manufacturer details for Nafion—there are multiple variants.

Define temperature control method: Was water bath or Joule heating used? Stability of temperature matters.

Describe gas measurement: How was SbH₃ generation confirmed or quantified—through GC or other analytical means?

Justify ultrasonic stirring: Provide rationale or reference showing why ultrasound is effective at high concentrations.

Control group needed: Was there a baseline group without electrolysis? Necessary to prove electrogeneration role.

Quantify hydrogen competition: Is hydrogen production rate measured or estimated to demonstrate its competitive relationship?

Explain decomposition threshold: Was the temperature decomposition of SbH₃ determined experimentally or inferred?

Add real sample test: Was this validated using actual industrial Sb wastewater? If not, suggest as future work.

Mechanism and Chemical Interpretation (21–30)

Thermodynamic support lacking: Consider calculating ΔG or Gibbs energy for SbH₃ formation/decomposition.

Phase change not explained: Elaborate whether Sb is recovered in crystalline, amorphous, or colloidal form.

Oxidation state unclear: Is Sb present as Sb(III) or Sb(V)? The valence state affects hydride generation.

Missing side-reactions: Was any stibine oxidized or lost via volatilization? Discuss gas-phase losses.

Clarify proton exchange role: Why does Nafion outperform CEM? Is it due to proton mobility, selectivity, or structure?

Incorporate reaction equations: Add half-cell or global electrochemical reactions for clarity.

Hydride safety: Mention the toxicity or handling precautions of SbH₃—very relevant for real-world applications.

Reversibility of SbH₃ reaction: Discuss whether SbH₃ can redissolve or react in aqueous phase.

Competing anions: Were any other ions (e.g., sulfate, nitrate) present? They may interfere in electrolysis.

Inclusion of pH dependency: Provide pH window in which SbH₃ formation is optimal.

Data Presentation and Analysis (31–40)

Add standard deviations: Were removal efficiency values repeated? Include standard error bars.

Statistical significance: Were differences between membranes or stirring methods statistically analyzed?

Include figures/tables: The abstract suggests comparative data—include bar or line graphs to visualize results.

Summarize orthogonal experiment: Present the L9 (3⁴) table or equivalent clearly and concisely.

Include power consumption analysis: Energy cost is critical for scale-up—present kWh/m³ values.

Define "highest removal efficiency" clearly: Is this for synthetic wastewater only or compared to real systems?

Optimize parameter interaction: Use contour or response surface plots if multiple variables affect removal.

Include control of ambient pressure: Was the system open or sealed? Could affect gas release dynamics.

Discuss reproducibility: State if results were consistent across batches.

Add detection methods: Specify ICP-OES, AAS, or other for final Sb quantification.

Writing and Grammar (41–45)

Fix grammar: “An optimized EML model were developed” → “was developed.”

Avoid sentence redundancy: Lines 14–15 and 23–24 repeat concept of removal being affected by competition.

Use formal tone: Avoid casual language like “notably accelerated.” Try “significantly enhanced.”

Check article usage: “a proton exchange membrane” not “employing proton exchange membrane.”

Simplify structure: Break long compound sentences for clarity, e.g., lines 22–24.

Scientific Significance and Journal Fit (46–50)

Highlight environmental relevance: Emphasize removal of a priority pollutant per EPA/EU regulations.

Add novelty statement: Explicitly state what is novel—electrogeneration of SbH₃? Use of SHG for Sb?

Discuss scale-up potential: Comment on feasibility in industrial wastewater treatment settings.

Propose future directions: Suggest exploring alternative membranes or hybrid ultrasonic-electrochemical reactors.

Align with journal aims: Frame the study within context of advanced oxidation/electrochemical wastewater remediation field.

Reviewers' comments:

Reviewer's Responses to Questions

**Comments to the Author**

1. Is the manuscript technically sound, and do the data support the conclusions?

Reviewer #1: Yes

2. Has the statistical analysis been performed appropriately and rigorously?

Reviewer #1: I Don't Know

3. Have the authors made all data underlying the findings in their manuscript fully available?

Reviewer #1: Yes

4. Is the manuscript presented in an intelligible fashion and written in standard English?

Reviewer #1: No

Reviewer #1: 1-The manuscript presents a novel application of electrochemical hydride generation (EHG) using a Nafion membrane for antimony (Sb) removal, which offers potential advantages over traditional electrochemical and adsorption methods. However, the novelty needs to be more explicitly emphasized. The authors should clearly differentiate their approach from other electrochemical techniques, such as electrocoagulation, and elaborate on how the use of Nafion and the recovery of Sb via SbH₃ decomposition significantly advances the field.

2-The experimental methods are generally well-structured, but important details are missing that may hinder reproducibility. Key parameters such as the spacing between electrodes, exact volume of electrolyte, electrode immersion depth, and duration of each test run should be clearly specified. Including a schematic diagram of the experimental setup with precise dimensions would improve clarity and help readers replicate the experiments.

3-The proposed removal mechanism based on SbH₃ formation and decomposition is reasonable, but relies heavily on indirect observations (e.g., silver mirror effect and post-experiment nitric acid soaking). To strengthen the evidence, the authors should consider incorporating direct or in situ measurements of gaseous SbH₃ (e.g., GC-MS, FTIR) in future work. Additionally, a discussion of potential safety hazards associated with SbH₃ and how they are mitigated would be valuable.

4-The extension of the method to real leachate from the Banpo antimony mine is a strong point. However, this section is relatively brief. The manuscript would benefit from a more comprehensive comparison between the treatment of synthetic and real wastewater, including how matrix complexity, oxidation states, and competing ions affect Sb removal. A summary table comparing removal efficiencies and pH effects in synthetic versus real samples would be a helpful addition.

5-While the manuscript is generally readable, the text in the Results and Discussion sections can be overly detailed and sometimes repetitive. The authors are encouraged to condense descriptions where possible and improve the logical flow. Subheadings or bullet points could help in summarizing key findings. Minor grammatical corrections and sentence restructuring are also recommended throughout the manuscript for improved clarity and professionalism.

**Do you want your identity to be public for this peer review?** For information about this choice, including consent withdrawal, please see our Privacy Policy

Reviewer #1: **Yes: ** Akbar Abbasi

---

## [Author Response · Author response to Decision Letter 1]

7 Aug 2025

1. Response to editor’s comments

Abstract Structure & Scientific Clarity (1–10)

1. Lack of clarity in the problem statement: The abstract does not clearly state why antimony removal is an urgent or novel topic. Please briefly introduce the environmental risk.

Reply: We have incorporated discussions regarding the environmental risks associated with antimony into the Abstract section (Line 13-14).

2. Improve technical flow: The transition between electrolysis, SbH3 formation, and Sb recovery feels abrupt—provide a clearer cause-effect structure.

Reply: We have restructured the descriptions of the electrolysis, SbH3 formation, and antimony (Sb) recovery processes in both the Abstract (Line 16-18) and the Methods section (Line 124-130).

3. Quantify the initial concentration ranges: State the initial Sb concentrations used in the experiments for context.

Reply: The initial concentration of antimony (Sb) in the synthetic wastewater (5 mg/L) has now been incorporated into the Abstract section (Line 18).

4. Avoid passive phrasing: Revise passive structures like “was employed” to active voice where possible.

Reply: We revised passive structures as suggested (Line 16).

5. Explain recovery vs. removal distinction: Is Sb merely removed or also recovered in solid form? Clarify in abstract.

Reply: The conversion of soluble antimony (Sb) to stibine gas (SbH3) enables its removal from water via volatilization. Subsequent heating of the liberated SbH3 induces decomposition into metallic antimony within a quartz tube, offering a viable approach for Sb recovery. This mechanistic pathway has now been explicitly stated in the Abstract section (Line 16-18).

6. Clarify mechanisms: State more explicitly how SbH3 formation and decomposition enable Sb separation and recovery.

Reply: The reaction mechanism involving SbH3 generation and subsequent thermal decomposition was elaborated in Section 2.1 (Line 126-130).

7. Reword ambiguous phrasing: "Overflowed from the solution" is vague; suggest "was released as gas" or similar.

Reply: We appreciate the suggestion and have now systematically revised all similar issues throughout the manuscript.

8. Include brief methodological note: Mention reactor configuration or stirring methods used, even if in brief.

Reply: Thanks for your suggestion. The reactor description has been revised, and the Nafion membrane installation protocol has been supplemented (Line 19, Line 106-120; Fig 3). The stirring methodology has been detailed (Line 25, Line 168-174).

9. Summarize results more clearly: Mention the optimum conditions in one concise summary sentence.

Reply: Thanks for your suggestion. The optimal conditions have been summarized in one concise sentence in the abstract (Line 31-34).

10. Add significance: Conclude abstract with a statement on the broader implications for wastewater treatment.

Reply: Thank you for your nice comments on our article. The corresponding content has been add into the Abstract section (Line 34-37).

Technical Depth and Experimental Design (11–20)

11. Compare to existing methods: Clarify how this method compares to conventional chemical precipitation or adsorption approaches.

Reply: The key distinctions from conventional precipitation or adsorption approaches are twofold: (1) The removal mechanism differs fundamentally, wherein traditional methods rely on precipitate formation while our approach facilitates contaminant degassing; (2) Precipitate adsorption systems suffer from irrecoverable metal loss, whereas this methodology demonstrates feasible metal reclamation pathways. These comparative analyses have been thoroughly elucidated in the revised manuscript (Line 121-130).

12. Explain electrode selection: What electrode materials were used? Were they inert or catalytic?

Reply: The electrochemical configuration employs inert platinum electrodes for both anodic and cathodic processes in this apparatus (Line 112, 117).

13. Include membrane specification: Provide model or manufacturer details for Nafion—there are multiple variants.

Reply: This has been designated in the text (Line 19, 110).

14. Define temperature control method: Was water bath or Joule heating used? Stability of temperature matters.

Reply: The experiment uses a circulating water bath method for temperature control. The electrolytic cell was placed in a self-made circulating water cooling temperature control device, and the external circulating water temperature was set (Line 163-165).

15. Describe gas measurement: How was SbH₃ generation confirmed or quantified—through GC or other analytical means?

Reply: While direct measurement of SbH3 concentration was not conducted, we inferred the generation of SbH3 indirectly by quantifying the metallic antimony recovered through thermal processing. This allowed back-calculation of the antimony recovery rate. However, limitations including the decomposition kinetics during heating and partial incomplete release of SbH3 may result in slightly underestimated values compared to actual production. We therefore fully recognize the necessity of direct gas-phase detection. In future work, we will implement direct or in-situ measurement of SbH3 gas using techniques such as Gas Chromatography-Mass Spectrometry (GC-MS) and Fourier Transform Infrared Spectroscopy (FTIR).

16. Justify ultrasonic stirring: Provide rationale or reference showing why ultrasound is effective at high concentrations.

Reply: The observed efficacy at high concentrations was experimentally determined through kinetic analysis. During the initial reaction phase when antimony concentrations were elevated, ultrasonic irradiation significantly enhanced Sb removal (Line 300-301). However, in the terminal reaction phase where Sb concentrations approached a lower level, ultrasonic treatment did not further enhance removal efficiency compared to conventional stirring methods under the given experimental conditions (Line 314-316).

17. Control group needed: Was there a baseline group without electrolysis? Necessary to prove electrogeneration role.

Reply: We have documented the experimental results for the without electrolysis group in the manuscript (Line 219-220) and presented the corresponding dataset in Fig 4b.

18. Quantify hydrogen competition: Is hydrogen production rate measured or estimated to demonstrate its competitive relationship?

Reply: Although direct measurement of hydrogen evolution rate was not conducted, the competitive relationship is indirectly evidenced by the following experimental observations:

(1) When current increased from 0.1A to 0.5A, the current efficiency of the target reaction decreased significantly (Line 380-382), suggesting substantial current consumption by side reactions (primarily hydrogen evolution);

(2) The elevated energy consumption per unit output (Lines 384-386) likely reflects energy losses attributable to hydrogen evolution side reactions.

Mechanistically, simultaneous reduction reactions at the cathode involve competitive consumption of protons and electrons between:

Hydrogen evolution: H· + H⁺ + e⁻ → H₂

Stibine formation: Sb³⁺ + 3H⁺ + 6e⁻ → SbH3

where both H⁺ and Sb³⁺ species compete for cathodic electrons and Nafion-membrane-transported protons. The proton transport channels within the Nafion membrane may further favor preferential proton consumption by the hydrogen evolution reaction.

Regarding detection challenges, hydrogen gas generated during SbH3 decomposition complicates discrimination of hydrogen sources for competitive analysis. We emphasize our agreement on the necessity of direct gas-phase detection as a critical improvement for subsequent studies. We plan to implement real-time HER monitoring using online gas chromatography-mass spectrometry (OGC-MS) in partitioned experimental setups.

19. Explain decomposition threshold: Was the temperature decomposition of SbH₃ determined experimentally or inferred?

Reply: The potential decomposition temperature of SbH3 was extrapolated from prior literature (Line 274-277). In our removal experiments, temperature variations influence not only SbH3 decomposition but also impact its electrochemical formation and volatilization processes. The observed removal efficiency results from the convergence of multiple factors, with comprehensive evaluation establishing 30°C as the optimal removal temperature. Regarding decomposition in the thermal recovery unit, the operating temperature (�600 °C) substantially exceeds the threshold for rapid decomposition reported in literature (200°C). This order-of-magnitude difference supports the conclusion that near-complete SbH3 decomposition occurs in the recovery apparatus.

20. Add real sample test: Was this validated using actual industrial Sb wastewater? If not, suggest as future work.

Reply: We sincerely appreciate your valuable suggestions regarding our research. As a preliminary validation, we have conducted tests using industrial antimony-laden wastewater (tailings leachate), with relevant data presented in the section 3.5 (Line 394-426). In future work, we will further investigate the treatment efficacy across diverse categories of actual industrial wastewater.

Mechanism and Chemical Interpretation (21–30)

21. Thermodynamic support lacking: Consider calculating ΔG or Gibbs energy for SbH₃ formation/decomposition.

Reply: We have exhaustively searched relevant databases (e.g., NIST WebBook, PubChem) and literature sources (Tamaru, 1955; Gong et al., 1885; O'Neil, 2013; Chen et al., 2021) but were unable to locate reported values for the entropy change (ΔS) or equilibrium constant (K) pertaining to SbH3 decomposition. Regrettably, current technical limitations prevent experimental determination of these parameters, making accurate calculation of ΔG for SbH3 formation/decomposition infeasible.

However, experimental evidence from literature consistently demonstrates slow decomposition at ambient temperature and rapid decomposition into elemental antimony and hydrogen gas at 200°C (Lines 274-277). This behavior supports the thermodynamic inference that ΔG for SbH3 decomposition is negative at 298K, indicating spontaneous decomposition under standard conditions.

References:

Chen J L, Siepmann J I. Simulating vapor–liquid equilibria of PH3, AsH3, and SbH3 from first principles[J]. The Journal of Physical Chemistry C, 2021, 125(9): 5380-5385.

Gong, B.; Liu, Y; Tan, W; Yan, H. The temperature effect of reacting solution on the hydride generation and the stability of hydride. Chinese J. Anal. Chem. 1985, 673-676.

O'Neil, M.J. (ed.). The Merck Index - An Encyclopedia of Chemicals, Drugs, and Biologicals. Cambridge, UK: Royal Society of Chemistry, 2013.

Tamaru, K. The thermal decomposition of stibine. Journal of Physical Chemistry 1955, 59, 1084-1088.

22. Phase change not explained: Elaborate whether Sb is recovered in crystalline, amorphous, or colloidal form.

Reply: The antimony recovery manifests as solid-phase deposition forming an antimony mirror on quartz surfaces (Line 129, Line 224, S1 Fig), with acid dissolution confirming metallic antimony composition (Lines 225-226). Thermal decomposition presents a feasible reference approach for antimony recovery, though scalable implementation requires further system optimization for industrial deployment.

23. Oxidation state unclear: Is Sb present as Sb(III) or Sb(V)? The valence state affects hydride generation.

Reply: The Sb-containing wastewater (Sb(III)) to be treated was placed in the cathode chamber (Line 119). Wastewater was prepared using potassium antimony tartrate (KSbC4H4O7, purity ≥ 99%, Fluka, Germany) (Line 145-146).

24. Missing side-reactions: Was any stibine oxidized or lost via volatilization? Discuss gas-phase losses.

Reply: We wish to underscore that the electrolysis process produces substantial hydrogen gas, creating a protective reducing atmosphere within the quartz tube. Under these conditions, stibine (SbH3) decomposition proceeds without significant oxidative or volatilization losses. We hypothesize that factors such as the escape kinetics of SbH3 from the aqueous phase and its thermal decomposition rate may have adversely affected the Sb recovery efficiency, resulting in the observed moderate yield 71.8% ± 3.6% (Line 228-230).

25. Clarify proton exchange role: Why does Nafion outperform CEM? Is it due to proton mobility, selectivity, or structure?

Reply: The manuscript describes the distinctive proton-conducting properties of the Nafion membrane (Line 208-212), whereas SbH3 formation fundamentally requires Sb–H bond formation and exhibits pH-dependent behavior. Therefore, the proton-sieving selectivity and high proton conductivity of the Nafion membrane collectively contribute to electrolyte pH stabilization, thereby enhancing SbH3 generation efficiency within the weakly acidic range (pH 5.73–6.71) and consequently improving antimony removal efficacy (Line 239-242).

26. Incorporate reaction equations: Add half-cell or global electrochemical reactions for clarity.

Reply: The relevant reaction equations have been incorporated into the manuscript (Lines 127).

27. Hydride safety: Mention the toxicity or handling precautions of SbH₃—very relevant for real-world applications.

Reply: We have supplemented the manuscript with content addressing SbH3 toxicity and corresponding preventive measures (Line 419-421).

28. Reversibility of SbH₃ reaction: Discuss whether SbH₃ can redissolve or react in aqueous phase.

Reply: We have made a possibly reasonable inference in the manuscript according to the experimental phenomenon that further temperature increase causes the decrease in removal efficiency, combined with the characteristic that temperature rise accelerates SbH₃ decomposition. Specifically, the reaction: SbH₃ + 3H₂O → Sb(OH)₃ + 3H₂, causes Sb to return to the solution (Line 269-279).

29. Competing anions: Were any other ions (e.g., sulfate, nitrate) present? They may interfere in electrolysis.

 Reply: In this experiment, the simulated wastewater in the Nafion membrane electrolytic cell was prepared using ultrapure water, and the removal reaction occurs at the cathode without interference from other ions. The small amount of potassium sulfate added in traditional electrolysis devices to enhance conductivity would not participate in the electrolytic reactions in the current study. However, actual mine wastewater contains more relevant ions, which represents an important aspect for further research on experimental wastewater treatment.

30. Inclusion of pH dependency: Provide pH window in which SbH₃ formation is optimal.

Reply: This study found that traditional electrolytic cells cause a significant increase in the pH of the catholyte, which affects the removal rate of Sb (i.e., reducing SbH3 generation). Therefore, when using a Nafion membrane, the pH remains stable (5.73–6.71), which is more favorable for SbH3 formation (Line 240).

Data Presentation and Analysis (31–40)

31. Add standard deviations: Were removal efficiency values repeated? Include standard error bars.

 Reply: All experiments were performed with triplicates, and error bars are included in all figures. For removal rates, standard deviations have been supplemented in the manuscript.

32. Statistical significance: Were differences between membranes or stirring methods statistically analyzed?

Reply: Thank you for pointing this out. We have now annotated significant differences in Figures 5 and 7.

33. Include figures/tables: The abstract suggests comparative data—include bar or line graphs to visualize results.

Reply: We appreciate your insightful suggestion regarding the presentation of EHG experimental data. We fully acknowledge that clear graphical representations are essential for effective result demonstration. As referenced in the Abstract, the comparative removal performance between traditional and Nafion membrane electrolysis device is presented in Section 3.1 through Fig. 4 (line chart showing temporal Sb variation trends under different configurations) and Fig. 5 (bar chart comparing final removal rates), with added statistical significance indicators. The temperature-dependent removal rate comparisons referenced in the Abstract are presented in Fig. 7 (bar chart, Section 3.2), w

---

## [Editor Report · Decision Letter 1]

12 Aug 2025

The Optimization of Electrochemical Hydride Generation Technology for Treating Antimony-Containing Wastewater

PONE-D-25-20648R1

Dear Dr. Liu,

We’re pleased to inform you that your manuscript has been judged scientifically suitable for publication and will be formally accepted for publication once it meets all outstanding technical requirements.

Kind regards,

Hailing Ma

Academic Editor

PLOS ONE

Additional Editor Comments (optional):

The authors have thoroughly addressed the reviewers’ comments and substantially improved the manuscript; I recommend acceptance for publication.
---

## [Editor Report · Acceptance letter]

PONE-D-25-20648R1

PLOS ONE

Dear Dr. Liu,

I'm pleased to inform you that your manuscript has been deemed suitable for publication in PLOS ONE. Congratulations! Your manuscript is now being handed over to our production team.

Kind regards,

on behalf of

Dr. Hailing Ma

Academic Editor

PLOS ONE